# Fatty Acid Receptor CD36 Functions as a Surrogate Parameter for Lymph Node Metastasis in Oral Squamous Cell Carcinoma

**DOI:** 10.3390/cancers13164125

**Published:** 2021-08-17

**Authors:** Selgai Haidari, Matthias Tröltzsch, Thomas Knösel, Paris Liokatis, Anastasia Kasintsova, Marian Eberl, Florian Ortner, Sven Otto, Florian Fegg, Marko Boskov, Florian A. Probst

**Affiliations:** 1Department of Oral and Maxillofacial Surgery and Facial Plastic Surgery, University Hospital, 80336 Munich, Germany; matthias_troeltzsch@hotmail.com (M.T.); paris.liokatis@med.uni-muenchen.de (P.L.); Anastasia.ksn@t-online.de (A.K.); Florian.ortner@med.uni-muenchen.de (F.O.); f.fegg@med.uni-muenchen.de (F.F.); marko.boskov@med.uni-muenchen.de (M.B.); florian.probst@med.uni-muenchen.de (F.A.P.); 2Department of Pathology, Ludwig-Maximilians University of Munich, 80336 Munich, Germany; Thomas.Knoesel@med.uni-muenchen.de; 3Department of Sport and Health Sciences, Technical University of Munich, 81675 Munich, Germany; marian.eberl@tum.de; 4Department of Oral and Maxillofacial Surgery, Martin-Luther University Halle, 06120 Halle an der Saale, Germany; Sven.Otto@uk-halle.de

**Keywords:** oral squamous cell carcinoma, lymph node metastasis, CD36, prognostic factor

## Abstract

**Simple Summary:**

The frequent occurrence of occult cervical lymph node metastasis is still a therapeutic challenge in oral squamous cell carcinoma (OSCC) and represents a limiting factor in terms of survival. The common staging of malignancy is not precise enough to predict the development of lymph node metastases and additional prognostic factors are needed. In this study we show that CD36, a protein related to fatty acid metabolism, is expressed in OSCC and correlates with the occurrence of lymph node metastasis. CD36 may be useful as a specific parameter for lymph node metastasis and as a progression parameter for survival. Therefore, CD36 could be useful for risk stratification regarding lymph node metastasis in OSCC and, beyond that, CD36 could also be a possible therapeutic target in future.

**Abstract:**

Purpose: To investigate the expression pattern of CD36 in a patient population with oral squamous cell carcinoma (OSCC) and to correlate CD36 expression with clinical and histopathological parameters. The hypothesis was that CD36 expression correlates with the occurrence of lymph node metastasis. Methods: To address the study objectives, a retrospective cohort study was conducted. Study variables included demographic, histopathological and survival data. CD36 expression patterns were assessed by immunohistochemistry on tissue microarrays (TMA). Logistic regression analysis, survival analysis and Cox proportional hazards model were performed. Results: High CD36 expression correlated significantly with a higher T-status, grading and occurrence of lymph node metastasis. The logistic regression with binary N status as a dependent variable showed that high CD36 expression increased the chance for lymph node metastasis 45-fold (OR = 44.7, 95% CI: 10.0–316). Patients with high CD36 expression had lower probabilities of progression-free survival. CD36 had a small and non-significant independent influence on progression-free survival. Conclusions: CD36 is expressed in OSCC and correlates with tumor grading, T-status, and especially the occurrence of lymph node metastasis. CD36 may be useful for risk stratification regarding lymph node metastasis in OSCC.

## 1. Introduction

Oral squamous cell carcinoma (OSCC) is a common cancer in the head and neck region showing a very heterogeneous course of disease [1,2]. A significant proportion of patients suffer from cancer recurrence and lymph node metastasis [1,3,4]. For the assessment of prognosis and treatment planning, tumor staging according to TNM classification and UICC system is commonly used [5]. Lymph node metastasis is common and even 30–40% of the patients suffer from occult nodal metastasis [6,7]. Therefore, the standard of care in OSCC is usually a selective neck dissection or comprehensive neck dissection [8].

The frequent occurrence of occult metastases poses a major therapeutic challenge. Occult metastases are typically not visualized on imaging due to their size [6,7]. Even the common usage of PET-CT did not improve the therapeutic decision making [9]. They are then often only revealed in the pathological examination of the removed cervical lymph nodes. Markers that have a high correlation with the presence of lymph node metastases would be able to influence the therapeutic decision.

The extension and requirement of a neck dissection in UICC I or II cases, especially in the upper jaw, is debated [10,11]. The treatment decision is made using the TNM or UICC classification based on CT-based staging and clinical impression. However, these systems fail to take into consideration the biological heterogeneity of OSCC [12]. In order to improve therapy, we need to identify biomarkers that have prognostic value for early detection of lymph node metastasis [4].

Metastatic tumor cells differ from common cancer cells. They possess the ability of epithelial mesenchymal transition—a process in which they acquire stem cell-like properties and gain the ability of interstitial migration [13,14,15]. These are essential skills for tissue invasion and the development of metastasis. The stem cell-like properties also increase resistance to radio- and chemotherapy [16]. In addition to the ability to detach from the cell association, the capability to endure the passage in lymphatic vessels and the subsequent nesting in the lymph node are prerequisites for the development of lymph node metastasis [17]. For this process, the tumor cells need a sufficient supply of energy, which is, as recent studies show, provided by fatty acids [18,19,20,21]. Cancer cells show an altered metabolic activity and are in high demand for energy for obtaining and keeping their malignant potential [22]. CD36 is a scavenger receptor that plays a pivotal role in uptake of fatty acids [19]. In several carcinoma, such as breast cancer, ovarian cancer and pancreatic cancer, CD36 drives the progression of cancer cells [22,23,24]. Furthermore, an in vitro study has shown that CD36 plays a role in the fat uptake of gastric cancer cells [25]. Pascual et al. demonstrated that the inhibition of CD36 led to the absence of lymph node metastasis in an orthotopic OSCC mouse model, showing the dependency on fatty acids and CD36 for initiating lymph node metastasis [18].

Further studies of CD36 showed the involvement in tumor cell metabolism, metastasis, immune escape, and therapy resistance with respect to radio- and chemotherapy [26,27]. Metastasis requires changes of cell-cell and cell-matrix adhesion, activation of growth and survival signals. It has been shown that high CD36 in OSCC cells leads to reduced expression of E-cadherin and its binding molecule β-catenin, thereby enabling migration. It was also shown for OSCC cells that higher expression of CD36 lead to a higher proliferation rate and upregulation of PDGFRβ compared to low CD36 expressing cells [28]. CD36 is involved in the activation of cancer stem cells and epithelial-to-mesenchymal transition via TGF-ß [27]. Altogether, the cellular changes mentioned above lead to tumor growth, tissue invasion and therapy resistance.

Whereas in several malignancies a correlation between CD36 expression and poor survival has been shown, the impact of CD36 expression on the prognosis of OSCC has not been investigated yet. Furthermore, it is unknown whether CD36 expression is a suitable marker for lymph node metastasis.

The presence of lymph node metastases is the primary limiting factor in OSCC. Regardless of the radiological N status of the neck, a cervical lymph node dissection is usually performed. It is questionable whether all T1 and T2 tumors with an N0 or N1 neck will benefit from this extensive procedure. Here, as mentioned above, occult metastases that are not detected on imaging are a major challenge. Markers could help to extend or limit a cervical lymph node evacuation, as well as adjuvant therapies regarding the individual risk for the presence of neck lymph node metastases.

Because of the above-mentioned challenges, the aim of the study was to investigate the expression of CD36 in a patient population with oral squamous cell carcinoma with complete follow-up. In a second step, CD36 expression should be correlated with clinical and pathological parameters. The hypothesis was that CD36 expression correlates with the occurrence of lymph node metastasis.

## 2. Materials and Methods

### 2.1. Patient Cohort

We conducted a retrospective cohort study to investigate the CD36 expression and tumor staging in a sample of OSCC patients consecutively admitted to our clinic. Our analysis included patients over 18 years of age who were surgically treated with curative intent at the Department of Oral and Maxillofacial and Facial Plastic Surgery of the LMU Munich, between 1 January 2010 and 31 December 2015. They had a primary manifestation of OSCC and a follow-up of at least 60 months.

Patients were excluded from the study if they had (i) a malignant disease in the head and neck region prior to the OSCC diagnosis, or (ii) a history of radiation in the head and neck area, or (iii) a history of chemotherapy or antibody therapy.

### 2.2. Study Variables

The study data was obtained by reviewing patient records, pathological reports and by analyzing the immunohistochemistry staining. The following variables were assessed for the study:

Expression of CD36, tumor size (T), occurrence of lymph node metastasis (N), distant metastasis (M), histological grading, age, gender, recurrence, and progression-free survival. For grading and staging, we used the current World Health Organization (WHO) Classification of Tumors of the Head and Neck (2017) and the 7th edition of the UICC TNM classification [29]. TNM-stage in all patients was based on postoperative pathological classification. CD36 expression was defined as the primary predictor variable and N-status was defined as the outcome variable.

### 2.3. Immunohistochemistry

For all included patients, tissue micro arrays (TMAs) were taken. Each histological specimen included two punch biopsies at a size of 0.6 mm. Staining was performed with automated immunostainer (Ventana Benchmark XT autostainer) with the XT ultraView DAB Kit (Ventana Medical Systems, Oro Valley, Arizona, USA). An anti-CD36 mouse monoclonal antibody, clone OTI3F4 (Origene Technologies GmbH, Herford, Germany), was used for staining. The stainings were performed according to the manufacturer manual and standardized protocol (Table 1). Squamous cell carcinoma of the tonsil served as a control.

A pathology specialist performed the evaluation of all specimens in a blinded manner. CD36 expression was categorized as follows: 0, none; 1, weak; 2, moderate; 3, strong. An intensity of 2 and 3 was considered as positive staining and therefore defined as overexpression of CD36. Intensity grades 0 and 1 were defined as low CD36 expression and 2 and 3 as high CD36 expression in the following (Figure 1). IHC staining stained the cell membrane. The staining was homogeneously distributed. The staining intensity was classified as above.

### 2.4. Statistical Analysis

Patient characteristics are presented stratified by CD36 status. High CD36 expression was defined as moderate or strong staining of the immunoassays, and low CD36 expression for patients with none or weak staining.

To analyze the association between high CD36 expression and N stage, we created a cross-tabulation, calculated sensitivity, specificity, PPV and NPV of CD36 as a single predictor for the occurrence of lymph node metastasis (N0 vs. N1/N2/N3). Moreover, we conducted a logistic regression analysis adjusting for age as a linear predictor and sex, T stage and grading as categorical variables (Model N1). CD36 was coded as a binary predictor (low vs. high) and N stage as a binary outcome variable with 0 = N0 stage (absence of lymph node metastasis) and 1 = N1 stage or higher. As sensitivity analysis, we also coded N stage as a categorical variable of four stages and used an ordinal logistic regression model (Model N2).

To evaluate the progression-free survival for different CD36 status, we created a Kaplan–Meier plot using R-Studio.

To analyze whether high CD36 expression had an impact on progression-free survival after surgery (counting any documented recurrence or death as an event), we applied a Cox proportional hazards model with age, sex, T stage, N stage and grading as additional covariates.

In order to corroborate the hypothesis that the CD36 pathway is predominantly via initiation of lymph node metastasis, we conducted another sensitivity analysis where we used CD36 in an ordinal regression model to predict the categorical grading variable using the same covariates as in Model N2.

All analyses were conducted using R statistical software, version 4.0.3 (R Core Team (2020). R: A language and environment for statistical computing. R Foundation for Statistical Computing, Vienna, Austria, https://www.R-project.org/). Logistic regression models were calculated by generalized linear models (GLM) using the binomial family. The ordinal logistic regression models were calculated with the polr function of the MASS package [30].We did not adjust for multiple testing.

## 3. Results

### 3.1. Description of Study Population

The patient cohort consisted of 83 patients (female *n* = 33, male *n* = 50; age 64.1 (mean) ± 16.5 (SD) years) with primary diagnosed oral squamous cell carcinoma (OSCC). The majority of OSCCs were located at the alveolar process of the mandible (*n* = 29/83, 34.9%) and the floor of the mouth (*n* = 19/83; 22.9%). Other locations included the anterior 2/3 of the tongue (*n* = 15/83; 18.1%), the alveolar process of the maxilla and soft palate (*n* = 8/83; 9.6%), and the alveolar process of the maxilla and hard palate (*n* = 12/83; 14.5%). It was shown that 56.6% of patients were diagnosed with limited stage of disease (pT1 and pT2) and 43.4% with advanced stage of disease (pT3 and pT4). Cervical lymph node metastasis (pN1, pN2 and pN3) was histologically confirmed in 40 patients (48.2%). Histopathological grading showed that OSCCs were in 16 of 83 patients (19.3%) well-differentiated (G1), in 52 of 83 patients (62.6%) intermediately differentiated (G2), and in 15 of 83 patients (18.1%) poorly differentiated (G3). Five of eighty-three (6%) patients exhibited distant metastasis. High CD36 expression within the specimens was evident in 38 of 83 patients (45.8%). The complete demographic, clinical and pathologic data are listed in Table 2.

### 3.2. Association between CD36 Expression and Lymph Node Metastasis

Of 40 patients suffering from lymph node metastasis, 32 (80%) had high CD36 expression (Table 3). In contrast, only 6 of 43 (13%) patients with a pN0 had high CD36 expression. The sensitivity of CD36 for pN+ status was 80% and the specificity was 86%. The positive predictive value of high CD36 status for lymph node metastasis was 84% and the negative predictive value of low CD36 for N0 status was 82%. Characteristics marked with * are significant with a *p* < 0.05.

The logistic regression with binary N status (pN0 vs. pN1–3) as the dependent variable, showed that after adjusting for sex, age, pT status and grading, high CD36 expression increased the chance for lymph node metastasis 45-fold (OR = 44.7, 95% CI: 10.0–316) (Table 4. Both CD36 expression and tumor grading were independent factors increasing chances for metastasis (for G2 OR = 3.17 and for G3 OR = 47.3).

### 3.3. Survival Analysis

An analysis of progression-free survival using the Kaplan–Meier method is shown in Figure 2. Patients with high CD36 expression had lower probabilities of progression-free survival. After 12 months of follow up, 87% (*n* = 39) in the low CD36 group were progression-free, whereas only 71% (*n* = 29) in the high CD36 group had a positive outcome. After 24 months, the discrepancy of progression-free survival increased to 82% (*n* = 38) in the low CD36 group and only 47% (*n* = 18) in the high CD36 group. At the end of follow up, 31 of 45 (69%) of patients with low CD36 expression were alive and recurrence-free, compared to only 13 of 38 (34%) of patients with high CD36 expression.

In the Cox regression model, we included the predictors age, sex, pT, pN and grading and observed that, adjusted for these covariates, the CD36 status had only a small and non-significant independent influence on progression-free survival, with high CD36 expression increasing the hazards for a negative outcome by 35% in our sample (HR = 1.35, 95% CI: 0.55–3.35, *p* = 0.5) (Table 5). The most important predictors for progression-free survival were pN, sex and age. N1 increased the hazards of a negative outcome by 142%, N2 by 342%, and N3 by 712%, compared to no lymph node metastasis (HR = 8.12, 95% CI: 1.67–39.4). Males had a 2.6-fold risk compared to females (HR = 2.59, 95% CI: 1.15–5.86), and every additional year of age increased the risk for recurrence or death by 5% (HR = 1.05, 95% CI: 1.02–1.08).

An analysis of progression-free survival of the N+ patients using the Kaplan–Meier method is shown in Figure 3. All patients included suffered from lymph node metastasis. Patients with high CD36 expression had lower probabilities of progression-free survival. Only eight patients with low CD36 expression suffered from lymph node metastasis compared to 32 with high CD36 expression. Due to (*n* = 8), a Cox regression was not performed. The *p*-value was 0.4 and thus the difference was not statistically significant. Higher case numbers are needed to make a statement, but at least a trend is apparent.

Finally, we investigated whether CD36 expression affects N+ status in T1 and T2 tumors. Table 6 shows that with high CD36 expression, lymph node metastases are present in 88% of cases.

## 4. Discussion

The frequent occurrence of metastasis in OSCC, as well as the presence of occult metastasis, poses a therapeutic challenge and is the limiting factor for patient survival [7,12]. The conventional grading of malignancy is not a precise enough tool to predict the development of lymph node metastasis. Preoperative staging investigations with CT or MRI typically fail to visualize occult metastases. Typically, these do not become apparent until histologic examination of the lymph node specimen. However, because only a fraction of the more than 200 lymph nodes in the neck are removed, these patients generally have a significantly higher risk for incidentally leaving affected lymph nodes in place. Here, additional indicators that correlate with the occurrence of lymph node metastases are helpful in order to perform therapy that is adjusted to the individual risk profile.

CD36 expression occurs in 45% of the OSCC cases in our cohort. This is an average amount compared with other entities. Published studies present a range of CD36 expression, from 19% in squamous cell carcinoma of the esophagus to 70% in cervical carcinoma [27,31].

The variations are explained by several factors: different IHC antibody and staining, different assessment, different sampling strategies of study populations, and most importantly, different tumor entities

We performed a histopathological examination of CD36 expression in OSCC to investigate it as a potential individual risk factor for the occurrence of lymph node metastasis. In terms of sensitivity and specificity, CD36 showed a promising sensitivity (80%) and specificity (86%) in univariate prediction of lymph node metastasis. In OSCC, there are constellations where the question of the necessity or feasibility of a neck lymph node dissection arises, for example, a T1 or T2 tumor in the region of the alveolar process of the upper jaw [10,11,32]. T1 und T2 tumors with a high CD36 also showed a strong correlation with the existence of lymph node metastasis (Table 6). These cases should be considered even in radiological N0 necks for lymph node dissection. In contrast to current clinical studies that recommend watch and wait in these scenarios, in these constellations, CD36 expression could support decision-making.

In the low CD36 group, more than 82% presented with pN0, whereas in the high CD36 group, only 16% patients were free of metastases. Additionally, with regard to T-stage, we saw that 67% had a small tumor size (pT1 and pT2) in the low CD36 group, compared to only 44% in the high CD36 group. Tumors with high CD36 expression showed much higher dedifferentiation; a G1 grading occurred rarely with 5% of the cases, whereas it was more frequent in the low CD36 group, with 67% of cases. Summarizing our results shows that lymph node metastasis, high grading and increased T-status occurred much more frequently in the high CD36 group than for low CD36 status, an indication that CD36 could be associated with a more aggressive disease course.

In our study, overexpression of CD36 increases the risk of lymph node metastasis by 45-fold. Our data showed that CD36 is strongly associated with the occurrence of lymph node metastasis and has a stronger association with positive N-status than grading.

Even though CD36 expression correlated significantly in the log-rank analysis with a poorer progression-free survival, the multivariate analysis indicated that it is not an independent prognostic factor. This is most likely due to the high correlation with a positive N-status [31]. Within the N+ group, the progression-free survival differed between high and low CD36 expression, but there are only few cases with low CD36 expression and lymph node metastasis. Therefore, further investigations with higher case numbers are required.

There are several factors how CD36 might contribute to a higher malignancy. The influence of CD36 positive cells on the development and metastasis of OSCC has been demonstrated both in vivo in animal experiments and in vitro [18,28]. Cancer stem cell- like properties, e.g., migration and proliferation, have also been demonstrated for CD36 overexpressing cells [33,34]. Overall, CD36 appears to be associated with epithelial–mesenchymal transition. Further in vitro experiments showed that CD36 influences cell proliferation via Ki67 [28]. Taking this into consideration, it fits the correlation with the grading, the T-status, and the general aggressiveness. Similar findings have been shown for oesophageal squamous cell carcinoma [31].

Other carcinomas such as ovarian carcinoma or hepatocellular carcinoma have also shown an association between CD36 expression and metastasis formation [26]. However, the cause of this association is debated. Cellular migration and passage in blood or lymphatic vessels requires a continuous and high supply of energy [18]. Fatty acids (FA) became the focus of attention in recent studies, showing that prostate cancer proliferation is fueled by FA [19,35]. There also appears to be a link between FA and the EMT [27]. It has been shown for OSCC cell lines that FA induced the EMT via CD36 [28].

Overexpression of CD36 presumably provides the cells with a high amount of FA, and consequently energy, which tumor cells need for invasive growth and metastasis [25,36]. Correspondingly, Pascual et al. showed in mice that a strict diet led to the absence of lymph node metastasis [22]. Furthermore, they demonstrated that it had the same effect as a CD36 neutralizing antibody. To what extent these results can be transferred to humans remains unclear. These findings might indicate a role of pre- and post-therapeutic nutrition and the influence on disease progression. This could be relevant for patients with cancer in the digestive tract or head/neck region where, due to cachexia, high doses of FA for weight gain are admitted [31].

This study showed a clear correlation between CD36 expression and the occurrence of lymph node metastasis.

Considering the important data of Pascual et al., who see CD36 as the key player in the development of lymph node metastasis in OSCC, it remains to be discussed why patients with lymph node metastasis have low CD36 expression [18]. It is conceivable that tumors negative for CD36 in the TMAs express it in other areas. Furthermore, the expression may be more heterogeneous in the tumors than it appears in the TMAs.

The following shortcomings of the study need to be discussed:

First, our study sample is a selection of patients with severe courses of head and neck cancers, where pN+ was much more common than in other published studies [11,12]. Potentially, the association of CD36 and lymph node metastasis is stronger in our sample of severe cancer cases compared to all head and neck cancer patients.

Second, we have performed a cross-sectional analysis of CD36 and tumor stage. This carries the risk that the observed relationship between CD36 and N stage is expression of an unobserved cofounder causing both high CD36 level and lymph node metastasis.

Third, due to the limited sample size of our study, we saw that some categories of our covariates had very small case numbers. This increases the danger that regression estimators are unstable. Accordingly, the very high OR and CI estimates for high CD36 and lymph node metastasis might indicate an overfitting of multiple regression models.

Another weakness of the study is that the influence of individual uptake of FA could not be adequately assessed retrospectively. In this regard, a prospective study is necessary to investigate a possible relationship between FA uptake, CD36, and the occurrence of lymph node metastasis.

In summary, CD36 is useful as a specific parameter for lymph node metastasis.

## 5. Conclusions

CD36 is expressed in OSCC and correlates with tumor grading, T-status, and especially the occurrence of lymph node metastasis. CD36 could be used in the future for risk stratification concerning lymph node metastasis and might guide surgeons in the decision-making process of the necessity of cervical lymph node dissection, especially if there is uncertainty whether a cervical lymph node dissection is indicated.

## Figures and Tables

**Figure 1 cancers-13-04125-f001:**
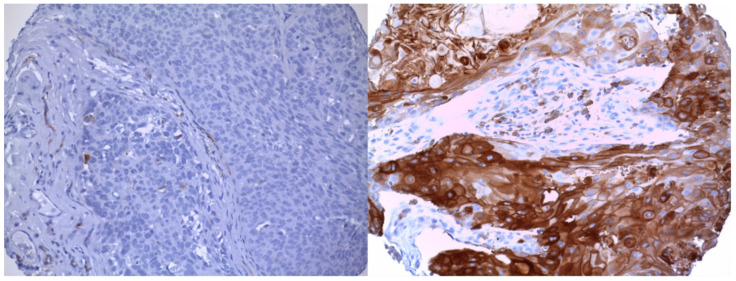
Tissue microarray stained for CD36. Representation of the immunoscoring classified; left low CD36 expression; right high CD36 expression. Magnification 20x.

**Figure 2 cancers-13-04125-f002:**
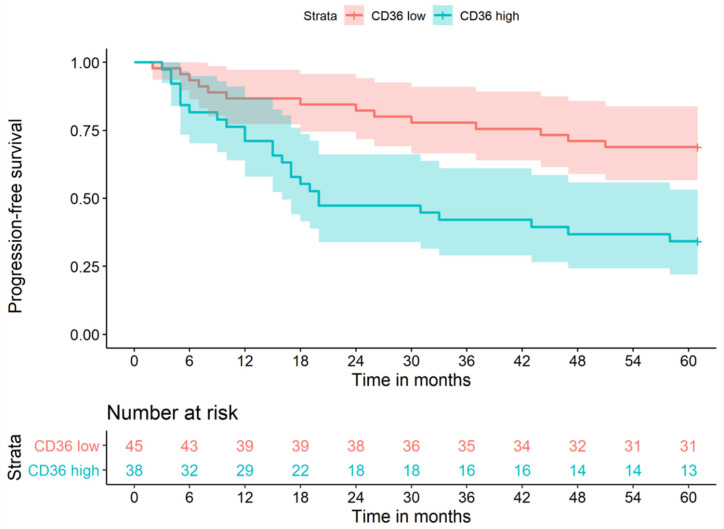
Progression-free survival low CD36 vs. high CD36 expression.

**Figure 3 cancers-13-04125-f003:**
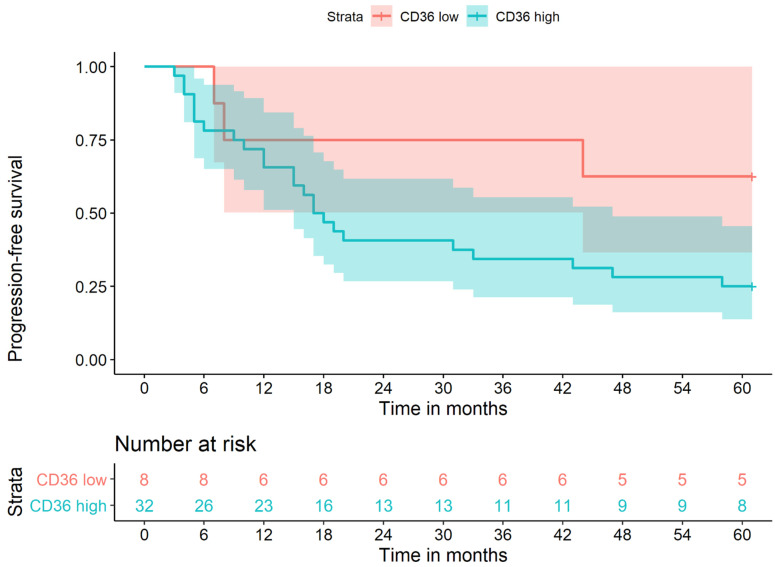
Progression-free survival in all N+ patients, low CD36 vs. high CD36 expression.

**Table 1 cancers-13-04125-t001:** Immunohistochemistry protocol.

Pretreatment	Heat Treatment with *p* ProTaqs EDTA Buffer 1 mM pH 8.0 (Fa.Quartett, 400500192)
Incubation	Incubation with primary antibody (CD36, (mouse monoclonal, clone OTI3F4)), 60 min RT, dilution 1:50
Detection system	ImmPRESS Anti-Mouse IgG Polymer Kit (Fa.Vector, MP-7402)
Chromogen	DAB + (Fa.Agilent Technologies, K3468)
Counterstaining	Hematoxylin Gill’s Formula (Fa.Vector, H-3401)

**Table 2 cancers-13-04125-t002:** Patient characteristics, diagnostic and outcome variables for study population.

Characteristic	*N*	Overall, *N* = 83	Low, *N* = 45	High, *N* = 38	*p*-Value
Age of patient (at time of surgery) [years]	83	64 (53, 78)	64 (52, 76)	66 (53, 81)	0.6
Sex	83				0.7
Female		33 (40%)	17 (38%)	16 (42%)	
Male		50 (60%)	28 (62%)	22 (58%)	
T classification	83				0.046
T1		28 (34%)	21 (47%)	7 (18%)	
T2		19 (23%)	9 (20%)	10 (26%)	
T3		12 (14%)	6 (13%)	6 (16%)	
T4		24 (29%)	9 (20%)	15 (39%)	
N classification	83				<0.001
N0		43 (52%)	37 (82%)	6 (16%)	
N1		19 (23%)	6 (13%)	13 (34%)	
N2		15 (18%)	2 (4.4%)	13 (34%)	
N3		6 (7.2%)	0 (0%)	6 (16%)	
M classification	83				0.017
M0		78 (94%)	45 (100%)	33 (87%)	
M1		5 (6.0%)	0 (0%)	5 (13%)	
Grading	83				0.005
G1		16 (19%)	14 (31%)	2 (5.3%)	
G2		52 (63%)	26 (58%)	26 (68%)	
G3		15 (18%)	5 (11%)	10 (26%)	

Table 2 presents patient characteristics, diagnostic and outcome variables for the study population stratified by CD36 expression. Significance refers to the correlation between the characteristic and CD36 expression.

**Table 3 cancers-13-04125-t003:** Cross-table showing the association between CD36 expression and lymph node metastasis.

		N Stage		Total
		N0	N+	
CD36	Low	37	8	45
	High	6	32	38
Total		43	40	83

N-status is assumed binary; chi-square, *p* < 0.001; Fisher’s exact test, *p* < 0.001; Spearman, 0.662 (*p* < 0.001).

**Table 4 cancers-13-04125-t004:** Results of the logistic regression model.

Characteristic	OR	95% CI	*p*-Value
CD36 expression			
Low			
High	44.7	10.0, 316	<0.001
Sex			
Female			
Male	2.61	0.66, 12.6	0.2
Age of patient (at time of surgery) [years]	0.99	0.94, 1.03	0.5
T classification			
T1			
T2	0.92	0.11, 6.51	>0.9
T3	0.57	0.06, 4.66	0.6
T4	0.36	0.05, 2.17	0.3
Grading			
G1			
G2	3.17	0.51, 28.6	0.2
G3	47.3	4.01, 1015	0.005

**Table 5 cancers-13-04125-t005:** Results of Cox regression model for progression-free survival.

Characteristic	HR	95% CI	*p*-Value
CD36 expression			
Low			
High	1.35	0.55, 3.35	0.5
Sex			
Female			
Male	2.59	1.15, 5.86	0.022
Age of patient (at time of surgery) [years]	1.05	1.02, 1.08	<0.001
T classification			
T1			
T2	1.20	0.41, 3.50	0.7
T3	2.36	0.82, 6.76	0.11
T4	1.92	0.72, 5.13	0.2
N classification			
N0			
N1	2.42	0.91, 6.42	0.076
N2	4.42	1.55, 12.6	0.006
N3	8.12	1.67, 39.4	0.009
Grading			
G1			
G2	1.54	0.52, 4.58	0.4
G3	0.99	0.26, 3.75	>0.9

**Table 6 cancers-13-04125-t006:** Cross-table showing the association between CD36 expression and lymph node metastasis in T1 and T2 Tumors.

		N Stage for T1 and T2		Total
		N0	N+	
CD36	Low	25	5	30
	High	2	15	17
Total		27	20	47

N-status is assumed binary; chi-square, *p* < 0.001; Fisher’s exact test, *p* < 0.001; Spearman, 0.662 (*p* < 0.001).

## Data Availability

The data presented in this study are available on request from the corresponding author.

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
