# Peer review of "Fatty Acid Receptor CD36 Functions as a Surrogate Parameter for Lymph Node Metastasis in Oral Squamous Cell Carcinoma"

_cancers, 2021, doi:10.3390/cancers13164125_

Round 1
Reviewer 1 Report
The manuscript has improved substantially after revision. Especially the importance of finding a good predictor of lymph node metastases has been highlighted.
Some minor comments:
1) The abstract still states that this is a cross-sectional study, please revise.
2) The cohort includes tumors of the soft palate which is an oropharyngeal location. These tumors should either be excluded or the authors need to argue why they have included these in a oral cancer cohort.
3) Foot note to Table 2 is misplaced.
4) Line 396-400: I see the authors’ argument in the response letter, but I still think this section would make better sense if you rephrased it: “First, our study sample is a selection of patients with severe course of head and neck cancers where pN+ was much more common than in other published studies [1112].”
5) Line 413-414: What is meant by “a progression marker for survival”? I suggest skipping the last part of the sentence or instead state that CD36 was not an independent marker of survival.
6) Minor linguistic corrections are needed, especially in the revised sections. Check for punctuations.
Author Response
Thank your for your kind words and your suggestions.
- Thank you we changed it. Line 25
-
Line 202
In principle, you are correct and it is not a tumor of the soft palate and an oral squamous cell carcinoma. The wording used is misleading. We have also looked at these 8 cases again in detail. These are carcinomas in the palatal region of the alveolar process that extend into the soft palate. Here we should have used the same designation as for "alveolar process of the maxilla and hard palate ". Therefore, we have adapted this and now refer to it as: "the alveolar process of the maxilla and soft palate ".
3. Thank you we changed that.
4. Thank you. For easier reading we followed your advise and changed the sentence according to your recommendation
5. Thank you very much for the hint. We changed it accordingly.
6. We did a spell check and corrected the errors. Thank you
Reviewer 2 Report
Thank you for your kind response that seems adequate to accept the article
Author Response
Thank your for your kind words. We did a spell check and hope that everything is now in order
Reviewer 3 Report
Comments to Cancers-1330771
This article investigated the correlation of CD36, a protein related to fatty acid metabolism, with clinicopathological features of OSCC. They used immunohistochemistry (IHC) to evaluate the expression intensity of CD36 in tumor tissue. And found that high expression of CD36 in tumor tissue correlated with lymph node metastasis and progression-free survival, but not overall survival.
The revised version was difficult to read. The author only answered part of my questions, and did not provide a point-to-point response. I suggest the author to respond to all my questions point-by-point.
My previous comments and author’s responses are listed below; please respond to the unanswered questions and my suggestions below.
Major comments:
- How was the study case number determined in the study?
Author response: Nil.
- Please provide p value in Table 1 (new Table 2) to show whether there are significant difference in CD36 expression between tumor or patient characteristics.
Author response: Only asterisk added.
My suggestion: Please define the meaning of asterisk in the footnote.
- In Table 4 (new Table 6), the Cox regression analysis of survival, it is not clear whether the survival was progression-free survival or overall survival.
Author response: Added progression-free on page 10, line 284.
My suggestion:
3.1. Please clearly indicate in the title of new Table 6 that it is a Cox regression analysis of factors related to “progression-free survival”.
3.2. All survival in the text should be specified. For example, on page 10, line 282.
3.3. In the conclusion (page 14), the correlation between CD36 with PFS was removed, probably because new Table 6 showed CD36 is not an independent factor of PFS. Therefore, the conclusion in the Abstract section should also be changed (page 1, line 34).
- It is obvious that N staging (N0-3) is the strongest determining factor of survival (New Table 6). Since CD36 and N staging is correlated, it is not appropriate to put N staging and CD36 expression simultaneously in the Cox regression model. To provide evidence that CD36 is an independent prognostic factor, which can provide clinical impact in addition to N staging. I suggest the author to perform a stratified analysis based on N staging. In other words, please analyze by Kaplan-Meir survival curve to show whether patients with CD36 low vs. high expression have different survival in patients with N0 status, and in patients with N+ status. If the author can prove that within the same N staging, CD36 provide additional information to predict progression-free survival, then it will have big impact. For example, according to CD36 expression status, we can decide whether extensive neck dissection should be done or whether post-operative chemotherapy should be added. Otherwise, N staging is more accurate than CD36, and CD36 will have little impact in clinical practice.
Author response: A Kaplan-Meir curve comparing the PFS in patients with N+, but different CD36 expression (low-CD36 vs. high-CD36) was added in new Fig. 3.
My suggestion: Please add patient numbers at risk under the Kaplan-Meir curve, and provide a p-value of comparison.
- In the comment 3, it is suggested to adjust age and gender in the stratified analysis, if they are also significant factors affecting progression-free survival. In addition, tumor progression in the primary site may have different mechanism to tumor progression in the LN or other distant site. Therefore, in the analysis of progression-free survival, progression in different sites (local or distant) may have to be analyzed separately.
Author response: See above. Thank you.
My suggestion: It is not clear where to see? Can you provide data about the site of disease progression?
- In the discussion, the author proposed that CD36 expression can predict LN metastasis in T1 and T2 tumor, and support for neck dissection. However, author should provide evidence to support the statement. Therefore, please do analysis to show that CD36 expression correlate with LN metastasis in tumors with T1/T2 stage.
Author response: A new Table 4 was added.
- Author Contributions and Institutional Review Board Statement are lacking.
Author response: We added that. Thank you!
Minor comments:
- The author stated that “CD36 had a small and non-significant independent influence on survival”. I suggested the author use the term “overall survival” to replace “survival” in the article. It will be more specific and understandable.
Author response: Changed that to progression-free survival.
- In the abstract, the author stated that “CD36 is expressed in OSCC and correlates with patient survival, tumor grading, T-status and especially the occurrence of lymph node metastasis”. However, no data regarding tumor grading and T-status were given in the abstract. Please add.
Author response: We added that briefly.
Author Response
1. A random number generator was used to select 100 patients from our internal tumor database within the specified time period. Of these, 17 were excluded because they were not primary oral squamous cell carcinomas.
2. We have now added the p-values as an additional column according to your suggestion. As a physician and scientist, I find this helpful and informative for the reader, as we are used to the kind of presentation in publications. We initially wanted to include the p-values in our first version. However, our co-author Marian Eberl, Master of epidemiology, has major concerns about the inclusion of p-values in this table and I would like to include them here:
"We discourage the use of p-values for Table 1. Recent discussions of the p-value have highlighted the limitations and outlined the recommended use within the context of hypothesis testing (Ronald L. Wasserstein & Nicole A. Lazar (2016) The ASA Statement on p-Values: Context, Process, and Purpose, The American Statistician, 70:2, 129-133, DOI: 10.1080/00031305.2016.1154108). Since Table 1 is intended for a description of the study population, we do not see that p-values would be used within the recommendations of the American Statistical Association."
Depending on your opinion, we would follow your advice and leave or remove the p-values.
3.
3.1. Thank you we changed that
3.2 We have now consistently used progression-free survival. I assume that you can now see this via the mark up function in the manuscript
3.3. We "removed" survival from the conclusion. Line 34
4. The p-value was added (Line 297). Numbers at risk are now provided with the new figure. We changed it back to the previous layout.
5." Age and sex" adjustment: this has been done in the Cox regressions. We do not consider it useful to include this in the Kaplan-Meier curve, even if it is possible, because the presentation of co-variables greatly reduces readability.
We did not distinguish between recurrence in the oral cavity or in lymph nodes in the data collection. In principle, we could still collect this data subsequently. However, we believe that this would not yield any significant results, since the number of cases would be too small to find significant differences due to the differentiation of subgroups
This manuscript is a resubmission of an earlier submission. The following is a list of the peer review reports and author responses from that submission.
Round 1
Reviewer 1 Report
The manuscript “Fatty acid receptor CD36 functions as a surrogate parameter for lymph node metastasis in oral squamous cell carcinoma” by Haidari et al aims at investigating the expression pattern of CD36 in oral squamous cell carcinoma (OSCC) and its correlation with clinical and histopathological parameters. They conclude that CD36 expression in OSCC is correlated with patient survival, tumor grading, T-status and lymph node metastasis, and that the marker may be useful for risk stratification with regard to lymph node metastasis in OSCC.
Introduction,
Line 59-60: “For this process, the tumor cells need a sufficient supply of energy, which is mostly provided by fatty acids [17].”
This statement needs to be modified. Using a single reference for such a complex field as cancer metabolism is not appropriate. Cancer cells are generally believed to be highly dependent on glucose, although they may also have other sources of energy such as FA and amino acids.
Line 69-74: The paragraph tries to summarize multiple functions of CD36, but it over-simplifies and offers the reader no real insight into how CD36 functions. More details are needed or the paragraph may just as well be deleted.
Line 75-78: “Whereas in several malignancies a correlation between CD36 expression and poor survival has been shown, the impact of CD36 expression on the prognosis of OSCC remains unclear. Furthermore, it is unknown whether CD36 expression is suitable as a prognostic marker for lymph node metastasis.” Is the impact of CD36 expression on prognosis of OSCC unclear because of conflicting results of previous studies (if so references should be given), or unknown due to lack of previous studies (if so the authors should rather state that it is unknown)? The main finding and conclusion is that CD36 can be used to predict lymph node metastases. However, the authors have not given any information on how lymph node metastases are evaluated today (clinically, imaging, histopathology). If CD36 should be a useful marker, the authors need to provide arguments that the current methods for identifying lymph node metastases indeed need a supplement.
Materials and methods
Line 88: Is it correct to call this a cross-sectional study when there is a follow-up time of at least 60 months?
Line 95: “Some of the 95 included patients were also examined in a study by Troeltzsch et al [26].” The relevance of this information is not clear, I suggest to skip this line.
Line 101-102: How is the TNM status determined (clinical or pathological)?
Line 112-113: More details on the IHC procedure should be given so that it is possible for the readers to repeat (include information such as dillutions, blocking, pre-treatment, visualisations/secondary antibody).
Line 115-116: The scoring was based on the staining intensity. There is no description of the staining, it should be provided either here or in the result section, of importance is which cells were stained (and scored!) – only cancer cells? Where was the staining localized (membrane, cytoplasm, both)?, was the staining homogenous (if not, the percentage of positive cells should maybe have been taken into account in the scoring)?
Results:
Line 158-160: Cancers of the tongue may either be classified as oral cancers (the anterior 2/3 of the tongue/mobile tongue) or oropharyngeal cancers (base of the tongue). Due to the differences in risk factors and survival of oral and oropharyngeal cancers, these anatomical sites should be separated. Please specify if the tongue cancers were oral, oraopharyngeal or both. The soft palate is also classified as the oropharynx. If the authors want to keep oropharyngeal cancers in the study, this should be clearly stated, and it should be noted that this is a population of oral and oropharyngeal cancers.
Table 1: Age: Is it mean age or median, what are the numbers in parenthesis?
Information on treatment is missing. As a minimum, it should be stated if all patients were treated with curative intent, as this is highly relevant when performing survival analyses. In survival analyses patients receiving palliative treatment should be excluded.
Line 188-190: The relevance of this alternative model is unclear. I suppose that several groups in this model will be very small, and I suggest skipping this section unless it is made clearer why it is included.
Line 198-199: “An analysis of progression-free and overall survival using the Kaplan-Meier method 198 is shown in Figure 2”. In figure 2 there is only one survival plot and it is not clear whether it is overall survival or progression-free survival. Please correct.
Line 209-217: This paragraph contains too many details as the same information is also given in Table 4.
Discussion:
Line 237-239: “The variations are explained by several factors: Different IHC antibody and staining, different assessment, different sampling strategies of study populations and most importantly different tumor biology.” How do the authors know that the different tumor biology is the most important reason for differing results?
Line 240-247: “We performed a histopathological examination of CD36 expression in OSCC to investigate it as a potential individual risk factor for the occurrence of lymph node metastasis. In terms of sensitivity and specificity, CD36 showed a promising sensitivity (80% ) and specificity (86%) in univariate prediction of lymph node metastasis.. In OSCC, there are constellations where the question of the necessity or feasibility of a neck lymph node dissection arises, for example a T1 or T2 tumor in the region of the alveolar process of the upper jaw [9, 30] [10]. In these constellations, CD36 expression could support decision making»
This paragraph needs revision. To me it appears that lymph node metastases and CD36 staining was assessed at one time-point only, probably at the time of diagnosis or treatment/surgery. This does not allow an evaluation of CD36 as a risk factor for lymph node metastases. The data only allow analyses of their association or correlation. Furthermore, as stated above, to argue that CD36 expression can be used to predict N-status, current methods for determining the N-status (such as imaging) should be discussed (and their limitations highlighted).
Line 251-253: “Tumors with high CD36 expression showed much higher dedifferentiation, a G1 grading occurred rarely with 5% of the cases whereas it was with 67% 252 more frequent in CD36 low group.”
I might have missed it, but I can’t find this information in the result section. Reuslts that are included in the discussion should also be presented in the results (Table 1?).
Line 253-256: “Summarizing our results show that lymph node metastasis, high grading and increased T-status, occurred much more frequently in the high CD36 group than for low CD36 status, indication that CD36 could be associated with a more aggressive disease course.”
… or just as well a more advanced stage of disease (the tumors may have been diagnosed at a later stage).
Line 263-271: here the authors discuss that CD36 overexpressing cells may harbor cancer stem cell properties. Cancer stem cells are believed to make up a small fraction of cancer cells, whereas CD36 staining in the OSCCs seemed to be wide-spread according to Figure 1. Furthermore, cancer stem cells generally have a lower proliferation rate than other cancer cells, this does not fit well with the increased proliferation induced by CD36 (ref 23). Thus, the discussion in this paragraph needs to be better developed.
Line 296-297: "First, our study sample is a selection of patients with severe course of head and neck cancers that required surgical intervention." Surgical removal of the tumor is standard procedure for OSCC in most countries, also for small tumors, thus this sentence doesn not seem to make sense.
Line 316-314: This summary is misleading as CD36 was not an independent prognostic factor for survival.
Conclucions:
It should be made clear that CD36 was not an independent predictor of survival. Antibody therapy has not been mentioned elsewhere in the manuscript, thus the last sentence of the conclusion seems misplaced.
Reviewer 2 Report
Dear Authors, congratulations on your research. It is very interesting how your research outlines the relation between CD36 and the prognosis of OSCC and node metastasis. Of course the study is retrospectively based so it has intrinsic scientific limitations. My suggestion is to implement the study as prospective. It could be interesting to find if you can a relation between CD36 and depth of invasion or in general new TNM 8th edition parameters. Literature must be updated with other markers relation with node metastasis (IMP3, Podoplanin). Lastly, since literature is very wide on markers relation with PNI and prognosis, can you add some data of relation between CD36 expression and PNI to be up-to-date with current research in literature? Best regardsReviewer 3 Report
Comments to Cancers-1241530
This article investigated the correlation of CD36, a protein related to fatty acid metabolism, with clinicopathological features of OSCC. They used immunohistochemistry (IHC) to evaluate the expression intensity of CD36 in tumor tissue. And found that high expression of CD36 in tumor tissue correlated with lymph node metastasis and progression-free survival, but not overall survival. The article is well written and provides informative messages to physicians. I hope my suggestions can help improve the article:
Major comments:
- How was the study case number determined in the study?
- Please provide a p-value in Table 1 to show whether there is a significant difference in CD36 expression between tumor or patient characteristics.
- In the Cox regression analysis of survival (Table 4), it is unclear whether survival indicates progression-free survival or overall survival.
- N staging (N0-3) is the strongest determining factor of survival (Table 4). Since CD36 and N staging is correlated, it is inappropriate to put N staging and CD36 expression simultaneously in the Cox regression model. To provide evidence that CD36 is an independent prognostic factor, which can provide clinical impact in addition to N staging. I suggest the author perform a stratified analysis based on N staging. In other words, please analyze by Kaplan-Meir survival curve to show whether patients with CD36 low vs. high expression have different survival in patients with N0 status and patients with N+ status. If the author can prove that within the same N staging, CD36 provides additional information to predict progression-free survival, then it will have a big impact. For example, according to CD36 expression status, we can decide whether extensive neck dissection should be done or whether post-operative chemotherapy should be added. Otherwise, N staging is more accurate than CD36, and CD36 will have little impact on clinical practice.
- In comment 3, it is suggested to adjust age and gender in the stratified analysis, if they are also significant factors affecting progression-free survival. In addition, tumor progression in the primary site may have different mechanisms to tumor progression in the LN or other distant sites. Therefore, in the analysis of progression-free survival, progression in different sites (local or distant) may have to be analyzed separately.
- In the discussion, the author proposed that CD36 expression can predict LN metastasis in T1 and T2 tumors, and support neck dissection. However, the author should provide evidence to support the statement. Therefore, please analyze to show that CD36 expression correlates with LN metastasis in tumors with T1/T2 stage.
- Author Contributions and Institutional Review Board Statement are lacking.
Minor comments:
- The author stated that “CD36 had a small and non-significant independent influence on survival”. I suggested the author use the term “overall survival” to replace “survival” in the article. It will be more specific and understandable.
- In the abstract, the author stated that “CD36 is expressed in OSCC and 32 correlates with patient survival, tumor grading, T-status and especially the occurrence of lymph 33 node metastasis”. However, no data regarding tumor grading and T-status were given in the abstract. Please add.